# Highly Sensitive Temperature Sensor Based on Vernier Effect Using a Sturdy Double-cavity Fiber Fabry-Perot Interferometer

**DOI:** 10.3390/polym15234567

**Published:** 2023-11-29

**Authors:** Miguel Á. Ramírez-Hernández, Monserrat Alonso-Murias, David Monzón-Hernández

**Affiliations:** Centro de Investigaciones en Óptica A. C., León 37150, Mexico; miguelrh@cio.mx (M.Á.R.-H.); monsealo@cio.mx (M.A.-M.)

**Keywords:** temperature-responsive polymers, fiber optic Fabry-Perot interferometer, the Vernier effect, temperature sensing

## Abstract

Temperature measuring is a daily procedure carried out worldwide in practically all environments of human activity, but it takes particular relevance in industrial, scientific, medical, and food processing and production areas. The characteristics and performance of the temperature sensors required for such a large universe of applications have opened the opportunity for a comprehensive range of technologies and architectures capable of fulfilling the sensitivity, resolution, dynamic range, and response time demanded. In this work, a highly sensitive fiber optic temperature sensor based on a double-cavity Fabry-Perot interferometer (DCFPI) is proposed and demonstrated. Taking advantage of the Vernier effect, we demonstrate that it is possible to improve the temperature sensitivity exhibited by the polymer-capped fiber Fabry-Perot interferometer (PCFPI) up to 39.8 nm/°C. The DCFPI is sturdy, reconfigured, and simple to fabricate, consisting of a semi-spherical polymer cap added to the surface of the ferrule of a commercial single-mode fiber connector (SMF FC/PC) placed in front of a mirror at a proper distance. The length of the air cavity (*L_air_*) was adjusted to equal the thickness of the polymer cap (*L_pol_*) plus a distance *δ* to generate the most convenient Vernier effect spectrum. The DCFPI was packaged in a machined, movable mount that allows the adjustment of the air cavity length easily but also protects the polymer cap and simplifies the manipulation of the sensor head.

## 1. Introduction

Fiber optic-based temperature sensors (FOTS) are a relatively new technology that has gained attention since it has been demonstrated that they can reach the competitive levels of sensitivity, response time, dynamic range, or resolution demanded by strategic sectors such as healthcare, automotive, chemical, oil and gas industry, and energy and power production. The size and lightweight, the immunity against electromagnetic external noise, the low chemical reactivity, and the biocompatibility of the glass used to fabricate the fiber optics have also contributed to the continuous growth of FOTS in the huge and well-established market of temperature sensor technologies [1]. There is a well-known preponderancy for fiber optic distributed sensors over point sensors for temperature measurement, however, for some applications where the size and location of the sensing zone are a concern, point sensors are the suitable or even the only solution.

Fiber optic Bragg grating is recognized as the point FOTS prototype; by the appropriate characterization process, it is possible to establish a one-to-one relationship between the temperature and the wavelength of the fiber Bragg grating reflection peak. This successful technology has overcome the low-temperature sensitivity of silica by taking advantage of the narrow reflection peak and the efficient and accurate wavelength shift interrogation methods developed to follow the wavelength position of the reflection peak. FOTS, like most sensing technologies, must deal with the compromise between sensitivity and dynamic range. Silica-based FOTS have demonstrated a very competitive temperature dynamic range from negative temperatures to hundreds of Celsius degrees. Of course, the temperature resolution achieved with these sensors is greater than 1 °C. The strategy to improve the temperature resolution of FOTS beyond the Bragg gratings has centered on finding a way to overcome the low sensitivity of silica fibers. Three main approaches, or a combination of them, followed to improve the temperature sensitivity of FOTS, can be summarized as follows: the fabrication of fiber optic devices where highly sensitive optical phenomena, such as interference or plasmonic resonance, can be generated steadily using specialty fibers (hollow core fiber, photonic crystal fiber, or polarization maintaining fiber) [2,3,4]; the inclusion of materials with a high thermo-optic coefficient [5,6,7]; or more recently, the implementation of fiber optic structures or schemes to generate the optical Vernier effect [8,9,10], i.e., the combination of two interference spectrum with free spectral range (FSR) slightly different to produce a secondary scale envelope with larger FSR in the resulted spectrum. 

One of the most recent and successful approaches to assessing FOTS with improved temperature sensitivity involves using fiber optic interferometers coated with a polymer to enhance the optical phase difference of the beams involved [5,6,7,11,12,13,14]. The polymer-capped extrinsic fiber Fabry-Perot interferometer (PCFPI) is one of the simplest FOTS proposed so far; it is very easy to fabricate, especially when transparent photocuring polymers, such as Norland Optics Adhesive (NOA) or polydimethylsiloxane (PDMS), are used. The polymer attached to the fiber optic tip acts as the Fabry-Perot cavity; the thickness and refractive index of the polymer cavity are affected by temperature changes. Increments or decrements in temperature modify the phase difference of the interference beams, producing a red- or blue-shift of the interference spectra, respectively. The polymer response to temperature changes determines the temperature sensitivity of the PCFPI; therefore, the only way to enhance the sensitivity of the PCFPI is by choosing one with a high thermo-optic coefficient (TOC) or a high thermal expansion coefficient (TEC). Temperature sensitivities around 0.2 to 0.8 nm/°C using PCFPI-based FOTS have been reported [10,11,12,13]. In most of these FOTS, the improvement of temperature sensitivity and therefore, in the resolution, depended on the properties of the polymer used. The dynamic range is limited in most cases to a temperature interval of −10 to 80 °C by the characteristics of the thermo-sensitive material. One alternative recently exploited to improve the sensitivity of the PCFPI-based FOTS is related to the Vernier effect. OFTS based on the Vernier effect with temperature sensitivities of 11.93 nm/°C [15], 23.22 nm/°C [16], and 19.22 nm/°C [17] have been reported, which represent an increment of one or even two orders of magnitude. Most FOTS based on the Vernier effect proposed so far, which includes a polymer cavity, have been made directly over the SMF tip, so these sensors can be introduced in confined spaces; however, the multi-step fabrication process in the microscale is complicated and requires specialized equipment. The dynamic range of these highly sensitive interferometric FOTS is small (40–46 °C) [15], (32–50 °C) [16], or (41–44 °C) [17] since the wavelength shift of the reflected or transmitted spectrum is difficult to track for larger temperature intervals. Highly sensitive temperature sensors, despite their small dynamic range, are very appreciated in biomedical applications [18], structural health monitoring [19], and marine environment monitoring [20], to mention a few. It is evident that for some of these applications, the head of the FOTS must be packaged to make them more robust to resist hostile environments or the manipulation of untrained final users in order to extend the use lifetime. Packaging considerably increases the size of a fiber optic thermometer.

In this work, we propose a fiber optic temperature sensor based on the Vernier effect using a sturdy dual-cavity Fabry-Perot interferometer. To construct the PCFPI on the surface of the ferrule tip of a single-mode fiber (SMF) connector, the surface of the ferrule is put in contact with that of the pre-cured polymer previously poured in a small container. After the connector moved away from the polymer liquid surface, part of the material remained attached to the surface of the ferrule. A semispherical cap over the ferrule connector is generated after curing the polymer, following the procedure recommended by the producers. The connector, with the PCFPI in the tip, was then inserted into a machinery piece to be placed in front of a small mirror attached to the other extreme. The distance between the surface of the semispherical cap and the mirror was adjusted to be slightly different from that of the optical path of the polymer cavity. This small cavity difference generates a clear-cut envelope in the reflectance spectrum. The temperature changes induce a change in the optical path length of the polymer cavity due to the change in the refractive index and thickness of the polymer cap, which causes a displacement of the envelope in the reflectance spectrum. The temperature sensitivity achieved with the DCFPI, formed by the polymer and air cavity, was 39.8 nm/°C, approximately 69 times larger than that obtained with a simple PCFPI. This FOTS is very simple to construct; it can be adjusted for different polymer cap thicknesses; therefore, it is possible to modify the sensitivity and dynamic range when necessary.

## 2. Fundamentals

The scheme of the temperature sensor based on the Vernier effect using a polymer-caped fiber Fabry-Perot interferometer (PCFPI), proposed in this work, is represented in Figure 1. The beam exiting the SMF core through the surface $S_{1}$ is propagated through the polymer cavity and part is reflected back at the curved polymer surface $S_{2}$. The rest was transmitted to the air cavity and then reflected at the mirror surface $S_{3}$, the reflected beams are recoupled at the fiber core, where they interfere among themselves and with the beam reflected at the interface $S_{1}$.

The mathematical expression to describe the reflected intensity of the DCFPI is:(1)$$\begin{array}{l} I_{DCFPI}=[R_{1}+R_{2}{\left(1-R_{1}\right)}^{2}\eta_{1}^{2}+R_{3}{\left(1-R_{1}\right)}^{2}{\left(1-R_{2}\right)}^{2}\eta_{2}^{2}+2\sqrt{R_{1}R_{2}}\eta_{1}(1-\\ R_{1})\cos\left[\frac{2\pi }{\lambda }\left(2OPL_{pol}\right)\right]+2\sqrt{R_{1}R_{3}}\eta_{2}(1-R_{1})\left(1-R_{2}\right)\cos[\frac{2\pi }{\lambda }(2OPL_{pol}\\ +2OPL_{air})]+2\sqrt{R_{2}R_{3}}\eta_{1}\eta_{2}{\left(-R_{1}\right)}^{2}\left(1-R_{2}\right)\cos\left[\frac{2\pi }{\lambda }\left(2OPL_{air}\right)\right]]\;I_{0}, \end{array}$$
where $I_{0}$ is the intensity of the incident light in the fiber tip end-face surface $S_{1}$, λ is the wavelength of the light, $\eta_{1}$ and $\eta_{2}$ are the coupling coefficients of the beam reflected at the surface $S_{2}$ and $S_{3}$, respectively, as shown in Figure 1. Both coupling coefficients depend on the characteristics of the optical path followed by the interference beams and delimited by the three surface interfaces. If the two reflected faces are flat, the coupling coefficient can be expressed by the relation [21]:(2)$$\eta_{flat}=\frac{{\left(\pi n_{pol}w_{o}^{2}\right)}^{2}}{L^{2}\lambda^{2}+{\left(\pi n_{pol}w_{o}^{2}\right)}^{2}}.$$

Nevertheless, since the $S_{2}$ surface has a concave shape, the calculus of the power coupling coefficient is through its general transfer matrix *M*, for any optical system, obtained by multiplying the corresponding ABCD matrices of each one of the n number of optical elements in the form: $M=M_{n}M_{n-1}M_{n-2}\cdots M_{3}M_{2}M_{1}$. This is based on the complex beam parameter method, also known as the ABCD Ray Matrix method, developed by Kogelnik et al. [22]. Thus, the general coupling coefficient depends on the transfer matrix and the confocal distance [23], as follows:(3)$$\eta_{general}=\frac{4z_{c}^{2}}{B^{2}+z_{c}^{2}\left(A^{2}+D^{2}+2\right)+z_{c}^{4}C^{2}}.$$

For the DCFPI, the coupling coefficients can be obtained from the coupling matrices:(4)$$M_{\eta_{1}}=M_{5}M_{4}M_{3}M_{2}M_{1}.$$
(5)$$M_{\eta_{2}}={M^{\prime }}_{9}{M^{\prime }}_{8}{M^{\prime }}_{7}{M^{\prime }}_{6}{M^{\prime }}_{5}{M^{\prime }}_{4}{M^{\prime }}_{3}{M^{\prime }}_{2}{M^{\prime }}_{1}.$$
where $M_{1}=\left[\begin{array}{cc} 1 & 0\\ 0 & \frac{n_{\mathrm{fcore}}}{n_{\mathrm{pol}}} \end{array}\right]$, $M_{2}=\left[\begin{array}{cc} 1 & L_{\mathrm{pol}}\\ 0 & 1 \end{array}\right]$, $M_{3}=\left[\begin{array}{cc} 1 & 0\\ -\frac{2}{R} & 1 \end{array}\right]$, $M_{4}{=M}_{2}$, $M_{5}=\left[\begin{array}{cc} 1 & 0\\ 0 & \frac{n_{\mathrm{pol}}}{n_{\mathrm{fcore}}} \end{array}\right]$, ${M^{\prime }}_{1}=\left[\begin{array}{cc} 1 & 0\\ 0 & \frac{n_{fcore}}{n_{pol}} \end{array}\right]$, ${M^{\prime }}_{2}=M_{2}$, ${M^{\prime }}_{3}=\left[\begin{array}{cc} 1 & 0\\ \frac{n_{air}-n_{pol}}{n_{air}R} & \frac{n_{pol}}{n_{air}} \end{array}\right]$, ${M^{\prime }}_{4}=\left[\begin{array}{cc} 1 & L_{air}\\ 0 & 1 \end{array}\right]$, ${M^{\prime }}_{5}=\left[\begin{array}{cc} 1 & 0\\ 0 & 1 \end{array}\right]$, ${M^{\prime }}_{6}=M_{2}$, ${M^{\prime }}_{7}=\left[\begin{array}{cc} 1 & 0\\ \frac{n_{air}-n_{pol}}{n_{pol}R} & \frac{n_{air}}{n_{pol}} \end{array}\right]$, ${M^{\prime }}_{8}=M\prime_{4}$, and ${M^{\prime }}_{9}=\left[\begin{array}{cc} 1 & 0\\ 0 & \frac{n_{pol}}{n_{fcore}} \end{array}\right]$.

With $n_{fcore}$ as the refractive index of the fiber optic core, and R is the radius of curvature of the polymer cap calculated in terms of the polymer cavity length $L_{pol}$ and the diameter of the ferrule connector $d_{c}$:(6)$$R=\frac{L_{pol}}{2}+\frac{d_{c}^{2}}{8L_{pol}}.$$

Since the diameter of the ferrule connector does not change, $d_{c}$ is constant, and then R only depends on $L_{pol}$. The radius of curvature of the polymer cap is fundamental to calculating the coupling coefficient of a concave polymer surface, which is present in the elements of the reflection and transmission ray-transfer matrix $\left[\begin{array}{cc} 1 & 0\\ -2/R & 1 \end{array}\right]$ and $\left[\begin{array}{cc} 1 & 0\\ -\left(n_{2}-n_{1}\right)/n_{2}R & n_{1}/n_{2} \end{array}\right]$, respectively, for a spherical surface. If the surface is plane (R→∞), the reflection and transmission ray-transfer matrix for a plane surface is recovered, and the coupling coefficient can be calculated using Equation (2).

The optical path lengths (OPL) of the polymer and air cavity are described by $OPL_{pol}=n_{pol}L_{pol}$ and $OPL_{air}=n_{air}L_{air}$, respectively, and $OPL_{pol}+OPL_{air}=n_{pol}L_{pol}+n_{air}L_{air}$ is the OPL of the dual cavity. However, the refractive index of the polymer $n_{pol}$ and the length of the polymer cap $L_{pol}$ is modified by the temperature (T) effect as follows [24]:(7)$$n_{pol}\left(T\right)=n_{0}\left(T_{0}\right)+{\left(\frac{\partial n_{pol}}{\partial T}\right)}_{T=T_{0}}\left(T-T_{0}\right),$$
(8)$$L_{pol}\left(T\right)=L_{0}\left(T_{0}\right)\left[1+\alpha_{T}\left(T-T_{0}\right)\right],$$
where $\partial n_{pol}/\partial T$ is the thermo-optic coefficient (TOC) and $\alpha_{T}$ is the thermal expansion coefficient (TEC). For PDMS, $TOC=-4.5\times 10^{-4}\;{{}^{\circ}C}^{-1}$ and $TEC=9.6\times 10^{-4}\;{{}^{\circ}C}^{-1}$ [25].

## 3. Experimental Results

### 3.1. Fabrication of the PCFPI and the Temperature Characterization

The DCFPI proposed in this work consists of a semi-spherical PDMS cap added to the end-face of a single-mode fiber (SMF) FC/PC connector. An illustration of the device is shown in Figure 1. The PDMS was made with 10 portions of elastomer and 1 portion of curing agent [26]. The SMF FC/PC pigtail was first spliced to an SMF FC/APC pigtail connected to the sm125 interrogator from Micronoptics^®^ to interrogate the fabrication process of the PCFPI. The connector of the SMF FC/PC was fixed in a translation stage (NRT150/M, Thorlabs) to control the movements and to put the ferrule face in contact with the polymer. Then, the ferrule moved away, and the polymer adhered to the surface of the ferrule adopting a semi-spherical or dome shape at the tip. It was possible to know the polymer cap thickness by analyzing the optical spectrum obtained with the sm125 interrogator and calculating the FFT in real time using an automated program in LabView. The position of the peak in the FFT spectrum provides the optical path length of the polymer cavity, which is monitored during the fabrication process. Several PCFPI were fabricated with this procedure and the thickness of the polymer cap was very similar. Different strategies to remove small portions of the polymer to obtain a specific cavity length are currently under analysis. We observed that with these procedures, the thickness of the polymer cavity changes by approximately 0.02 mm. Due to the surface tension of the polymer on the face of the connector, each time a polymer portion is removed, it re-forms the shape of the concave surface. However, in this work, no changes in the physical cavity length were pursued. Finally, the polymer-capped connector was heated at 60 °C for 1.5 h using a Peltier plate (Orbital mixing chilling/heating dry bath, Ecotherm) for the curing of PDMS. When PDMS is cured, we proceed to characterize the temperature response of the PCFPI fabricated; a representation of the experimental set-up used is shown in Figure 2.

The PCFPI made over the connector tip was first exposed to temperature variations from 30 to 36 °C. Since the thermal transfer from the heated block to the polymer tip is contactless, it is mandatory to use a thermometer close to the PCFPI to obtain an accurate reading of the PDMS cap temperature. When the temperature is increased, the optical spectra of the PCFPI intensity reflection shift to larger wavelengths, as can be seen in the upper graph of Figure 3a. The wavelength position of one maximum of the interference pattern near 1550 nm was tracked, and these values (blue light dots) were used to construct the plot in Figure 3b. These experimental points are fitted by a line (blue light) with a slope of 0.54 nm/°C, which represents the temperature sensitivity of the PCFPI. When the temperature of the heater reached 36 °C, it was turned off, and then the temperature started to decrease. The spectrum of the intensity reflection of the PCFPI was recorded at different temperatures, and some of them are shown in the lower graphs of Figure 3a. The wavelength of one of the maxima for different temperatures is plotted as dark blue dots in Figure 3b; the slope of the linear fit (dark blue line) was 0.58 nm/°C which represents the temperature sensitivity of the PCFPI when the temperature was decreased. The temperature sensitivity of the device is very similar when the temperature is increased or decreased.

### 3.2. Simulation of the DCFPI and the Response to Temperature Changes

It has been demonstrated that the PCFPI has a good temperature sensitivity (0.58 nm/°C). We propose to construct a DCFPI to generate the Vernier effect as one method to enhance the PCFPI sensitivity. The DCFPI is formed when the PCFPI is placed in front of a reflective surface, as can be seen in Figure 1. To generate the Vernier effect, it is necessary to set the OPL of both cavities nearly equal, i.e., $OPL_{pol}\approx OPL_{air}$. The small difference in the length of the cavities can be expressed as $OPL_{air}=OPL_{pol}+\rho$, where $\rho \ll OPL_{pol}$. Based on the theoretical model described in Section 2, we proceed to simulate the reflection intensity of the DCFPI using the Equations (1)–(8) and considering the following parameters:$\;L_{pol}=0.58\;\mathrm{mm}$, $L_{air}=0.84\;\mathrm{mm}$, $n_{pol}=1.39$, R=1.127 mm, $R_{1}=0.0006$, $R_{2}=0.025$, $R_{3}=0.95$ and $n_{air}=1.00$. The coupling coefficients calculated using Equations (3)–(6) at a wavelength λ=1550 nm are $\eta_{1}=0.1027$ and $\eta_{2}=0.0544$. The small difference in the OPL of air and polymer cavities ρ was set equal to 30 μm since this value assures the observation of two nodes in the wavelength span from 1510 to 1590 nm, the emission band of the interrogator used in this work. The simulated spectra of the DCFPI assuming a temperature of 26.7 °C is shown in Figure 4a. It is important to mention that the length and refractive index of the polymer cavity is affected by the surrounding temperature according to the mathematical expressions (7) and (8) governed by the TOC and TEC of PDMS. In the spectrum shown in Figure 4a, it is possible to distinguish a series of visible envelopes that modulate the original spectrum of the PCFPI shown in Figure 3a. These envelopes are produced by the superposition of the interference fringes of the polymer and air cavities of the DCFPI and are associated with the Vernier effect. Two dotted red lines were added to the spectrum of Figure 4a to identify the wavelength of two consecutive nodes and labeled as $\lambda_{A}$ and $\lambda_{B}$. When we assume an increment of 0.1 °C in the ambient temperature, the OPL of the polymer cavity is changed, and the simulated spectrum shown in Figure 4b is red-shifted with respect to that of Figure 4a. The large displacement of $\lambda_{A}$ and $\lambda_{B}$ is due to the Vernier effect. No other change in the simulated spectrum obtained for a *T* = 26.8 °C compared with that obtained for *T* = 26.7 °C is observed.

### 3.3. Fabrication of DCFPI

The procedure to construct the DCFPI is illustrated in detail in Figure 5 and can be described as follows: first, the ferrule with the polymer cap (1) was inserted through the hole machined and fixed to threaded fiber adapted in the aluminum case (2), then a circular mirror (3) glued to the tip of a screw (4) was approached to the polymer cap by driving the screw through the nut machined in the other side of the case. These elements constitute the head of the DCFPI proposed in this work for temperature sensing. The length of the air cavity between the polymer cap and the mirror is adjusted by moving the screw. In the same way as the PCFPI fabrication process, the sm125 interrogator and its automated program were used to calculate the OPL of the air cavity in order to adjust the small difference ρ between the OPL of the air and polymer cavities and generate the Vernier effect. Since the thread of the screw is fine, it is possible to set the mirror in front of the polymer cap in an accurate way. The ρ value is directly related to the temperature sensitivity of the DCFPI sensor. In our case, ρ was set around 30 μm measured through the FFT; this is the most appropriate value to obtain the maximum sensitivity without losing the chance to track two nodes in each spectrum in the wavelength span from 1510 to 1590 nm. Using a light source with a broader span, it is possible to increase the sensitivity and dynamic range of this temperature sensor by reducing the value of ρ.

The experimental setup used to characterize the DCFPI is presented in the scheme of Figure 6. The DCFPI sensor head stands above a metal block in a Benchmark dry bath to set the temperature changes. A digital thermometer was placed near our DCFPI to monitor the temperature of the surroundings. The SMF was connected to a ENLIGHT, MicronOptics interrogator sm125 (Atlanta, GA, USA) to collect the reflection spectra. Before the start of the procedure to characterize the response of the DCFPI to temperature changes, we proceed to obtain the reflected spectrum at two temperatures, 26.7 and 26.8 °C, to compare them with that obtained by simulation. The spectra obtained using the experimental set-up represented in Figure 6 are shown in Figure 4c,d. In both cases, two red dashed lines are shown to mark the position of two nodes in the spectra. As can be seen, the shape of the experimental spectra and the position of the nodes are very similar to that of the simulated spectra shown in Figure 4a,b. Since the node shift of the experimental spectra corresponds in the same way with those simulated, it is possible to assume that the theoretical model proposed is a good approximation to describe the principle of operation of the DCFPI as a temperature sensor. A wavelength displacement of the nodes of around 4 nm when the surrounding temperature is increased by 0.1 °C allows us to estimate a sensitivity of 40 nm/°C. There is an increment of around two orders of magnitude in the temperature sensitivity of the DCFPI compared with that exhibited by the PCFPI.

### 3.4. Temperature Characterization of the DCFPI

In order to corroborate that the double cavity formed when the PCFPI is placed in front of the mirror, just as the set-up shown in Figure 5, can be used for temperature sensing with improved sensitivity by generating the Vernier effect, we proceed to the characterization of the DCFPI. The temperature of the dry bath was set to 35 °C, and the equipment was turned off. As the temperature decreased, the spectrum of the DCFPI was recorded, and the temperature assigned to each spectrum was exhibited by the digital thermometer placed on the side of the DCFPI. The optical spectrum shows a dense fringe pattern where two nodes can be identified (see Figure 7a, Figure 8a, Figure 9a and Figure 10a). The upper spectrum of Figure 7a was recorded at a temperature of 30.3 °C and exhibits two nodes labeled as Node 0 and Node 1 and located at wavelengths of around 1539 and 1579 nm, respectively. It can be seen that the envelope spectrum shifts toward shorter wavelengths as the temperature decreases. Within a temperature interval of 31.7 to 29.5 °C, Node 0 displaces along the span of wavelengths from 1510 to 1590 nm; the source emission of the MicronOptics interrogator sm125. Node 1 appears at a temperature of 30.6 °C and displaces through the whole wavelength span (Figure 8a); when a temperature of 28.5 °C is reached, it moves out of the wavelength interrogation span. Practically, at the temperature that Node 0 moves out of the wavelength frame, Node 2 appears on the opposite side at shorter wavelengths. At a temperature interval of around 29.5 to 27.5 °C, Node 1 and Node 2 can be followed. When Node 1 is displaced out of the screen Node 3 appears and shifts toward shorter wavelengths as the temperature decreases. Then, at a temperature of around 27.5 Node 2 disappears and appears Node 4. Tracking the wavelength position of the nodes through the frame, it is possible to plot the temperature characterization curve of nodes, as shown in Figure 7b, Figure 8b, Figure 9b and Figure 10b. It is complicated to track the displacement of a single node in a wide range of temperatures due to the high sensitivity of the Vernier effect. However, since two nodes are present in the optical spectra, both can be tracked by considering the overlap between the two characteristic curves. It was possible to follow 4 different nodes by the overlap between them (Figure 7, Figure 8, Figure 9 and Figure 10). Tracking the nodes’ displacement in the spectra could allow us to estimate the temperature variations but not to identify which node, which is related to a particular temperature range. For example, the position of the nodes at a temperature of 30.3 °C (Figure 7a) is very similar to that obtained at a temperature of 29.2 °C (Figure 8a), 28.1 °C (Figure 9a), and 27.1 °C (Figure 10a), but the FSR of the envelope is 40.3 nm, 41.5 nm, 42.4 nm, 43.4 nm, respectively. The FSR could help us to distinguish if we are following Node 1 or Node 3. The position of the nodes and the FSR of the envelope can be used to avoid the apparent ambiguity in the identification of the node and, in consequence, the temperature. Thus, a FOTS with a temperature dynamic range of 26.8 °C to 31.7 °C, as seen in Figure 11, was demonstrated. As the temperature decreases, the sensitivity of the device increases since the length of the PCFPI is closer to the length of the air cavity, and the free spectral range (FSR) of the envelope increases. This is more evident in the temperature range of 26.7 to 28.4 °C (Figure 10) when sensitivity increases to 39.8 nm/°C, which is 69.2 times the sensitivity of a conventional PCFPI temperature sensor.

Accordingly, it was demonstrated that the DCFPI proposed here significantly improves the temperature sensitivity of a PCFPI, compared with other FOTS based on the Vernier effect (see Table 1), this device exhibited a very competitive sensitivity, furthermore, the DCFPI possesses other interesting characteristics that could be attractive for some applications, for example, the simple fabrication process, the robustness provided by the connector, and the possibility to adjust the length of the air cavity by the movement of the mirror to select the sensitivity and dynamic range of the FOTS. The temperature stability of the sensor was also proved by tracking the wavelength position of a node in the spectra measured at 26.7 °C for 10 min, as seen in Figure 12. It shows that the node of the optical spectrum of the sensor had barely shifted around 1.2 nm over 10 min. Using the temperature sensitivity of 39.8 nm/°C, the slope of the linear fit of experimental data in Figure 10b, it is possible to estimate that the error in the temperature measurement is equal to or smaller than 0.04 °C. It is important to consider that the envelope dip is broad; this also contributes to the uncertainty in determining the displacement of the envelope dip (minimum). This can contribute to the deviation of the experimental data from the ideal line approximation of the calibration curves shown in Figure 7b, Figure 8b, Figure 9b and Figure 10b.

We are developing the test of a new DCFPI temperature sensor with slight structural variations to improve the temperature transference between the dry bath block and the head sensor. In addition, the use of a Hyperion si155 interrogation system with a wavelength span of 1460−1620 nm is considered for future tests. We estimate that these changes could be helpful to improve the sensitivity and dynamic range of the sensor discussed here.

## 4. Conclusions

We propose a simple-to-fabricate temperature sensor based on a modified extrinsic Fabry Perot interferometer made with a polymer-capped SMF FC/PC connector combined with the optical Vernier effect. Investigating its temperature performances through theoretical and experimental analysis, we demonstrate that it is possible to increase up to 69 times the temperature sensitivity of a conventional EFPI with a polymer cap. The structure of the sensor head is sturdy, and the possibility to adjust the OPL of the air cavity in a simple manner by just turning the screw to equal it to that of the polymer cavity of the PCFPI increases its versatility. We believe all these features make this proposal very attractive for applications where FOTS with high-temperature sensitivity, in a very narrow range, is required.

## Figures and Tables

**Figure 1 polymers-15-04567-f001:**
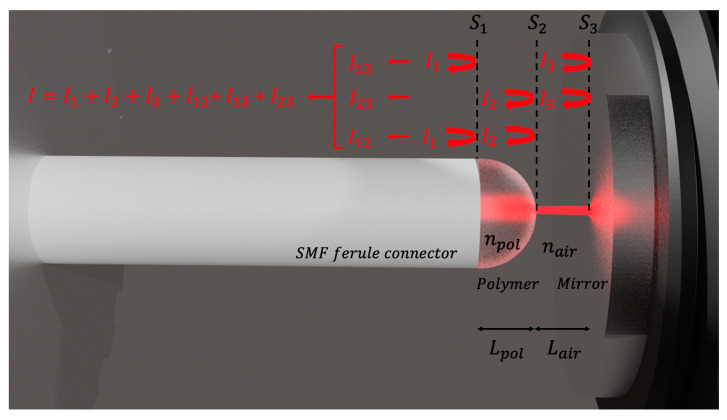
Internal view of the structure of the DCFPI-based temperature sensor.

**Figure 2 polymers-15-04567-f002:**
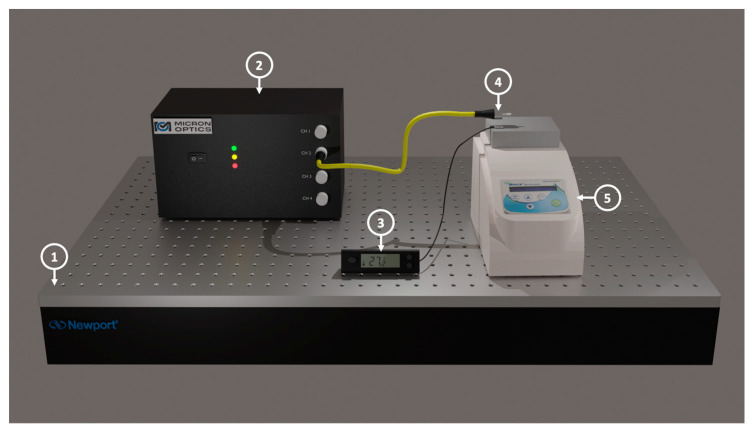
Experimental set-up used for temperature characterization: (1) optical table; (2) the sm125 interrogator; (3) digital thermometer with thermocouple; (4) ferrule connector with a polymer cap; (5) dry bath with block.

**Figure 3 polymers-15-04567-f003:**
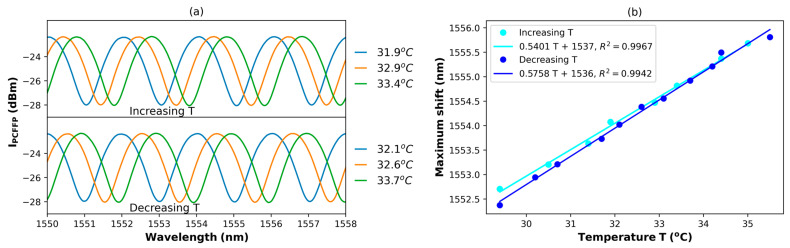
(**a**) Spectra of the measured $I_{PCFPI}$ and (**b**) the characteristic curve of the wavelength shift for a PCFPI with a PDMS cavity in a range of temperature of $30\;\mathrm{to}\;36{\;}^{o}C$.

**Figure 4 polymers-15-04567-f004:**
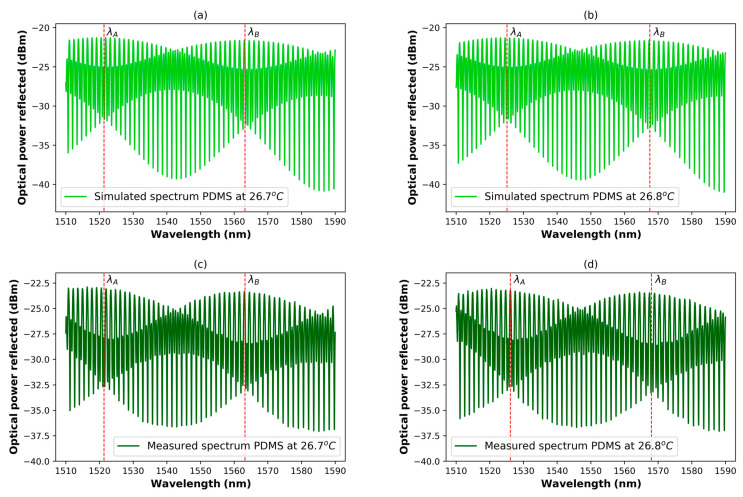
(**a**,**b**) Simulated and (**c**,**d**) experimental reflection spectra of a DCFPI when the surrounding temperature was $26.7{\;}^{o}C$ and $26.8{\;}^{o}C$, respectively.

**Figure 5 polymers-15-04567-f005:**
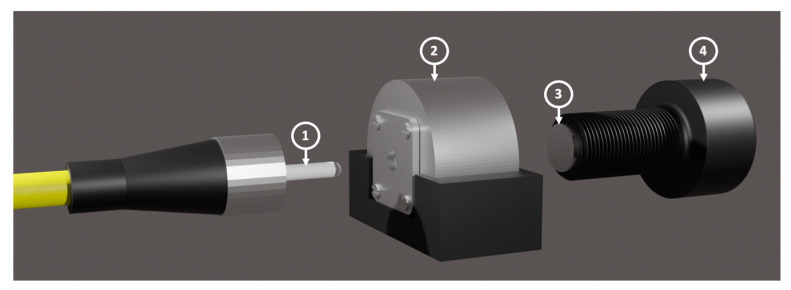
Components of the DCFPI sensor head: (1) Ferrule connector with a polymer cap; (2) Aluminum mount; (3) Plane mirror with a circular shape; (4) Fine thread screw.

**Figure 6 polymers-15-04567-f006:**
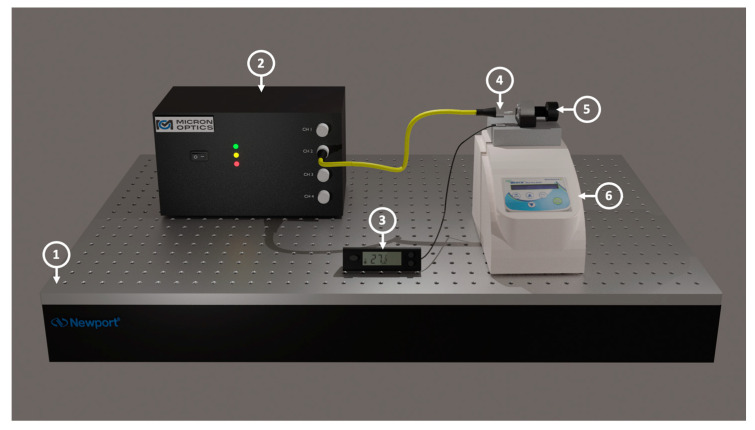
Experimental set-up used for temperature characterization of the DCFPI: (1) optical table; (2) the sm125 interrogator; (3) digital thermometer with thermocouple; (4) ferrule connector with a polymer cap; (5) screw with a mirror attached; (6) dry bath with block.

**Figure 7 polymers-15-04567-f007:**
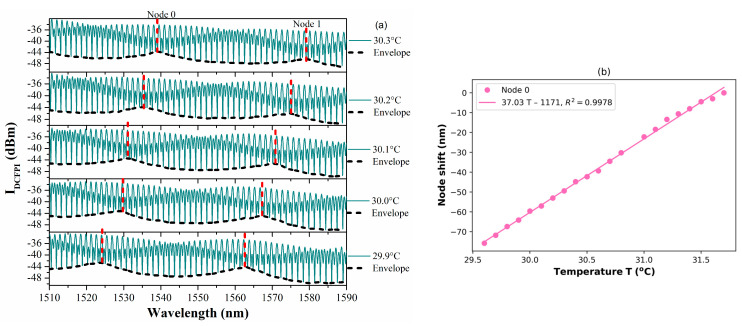
(**a**) Measured spectra of the DCFPI and (**b**) the characteristic curve of the node wavelength shift in the temperature range of 29.6 to 31.7 °C.

**Figure 8 polymers-15-04567-f008:**
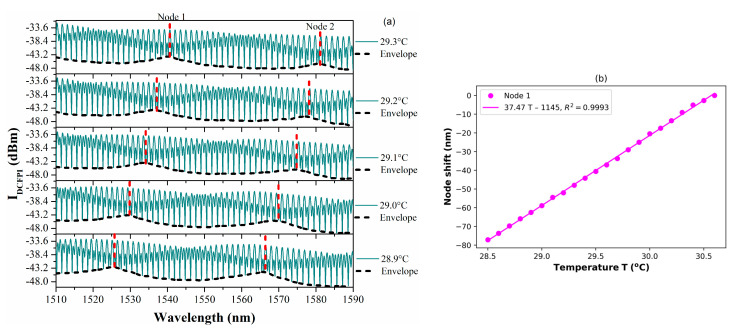
(**a**) Measured spectra of the DCFPI and (**b**) the characteristic curve of the node wavelength shift in the temperature range of 28.5 to 30.6 °C.

**Figure 9 polymers-15-04567-f009:**
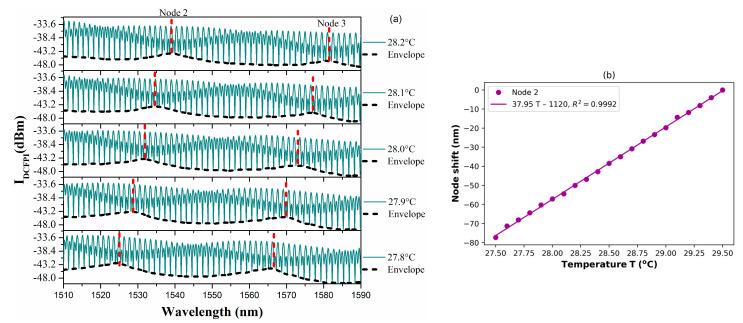
(**a**) Measured spectra of the DCFPI and (**b**) the characteristic curve of the node wavelength shift in the temperature range of 27.5 to 29.5 °C.

**Figure 10 polymers-15-04567-f010:**
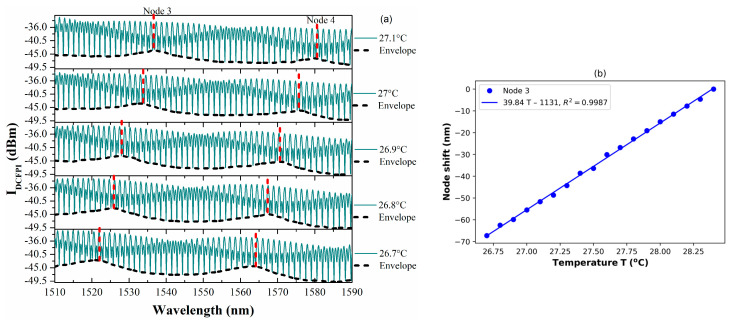
(**a**) Measured spectra of the DCFPI and (**b**) the characteristic curve of the node wavelength shift in the temperature range of 26.7 to 28.4 °C.

**Figure 11 polymers-15-04567-f011:**
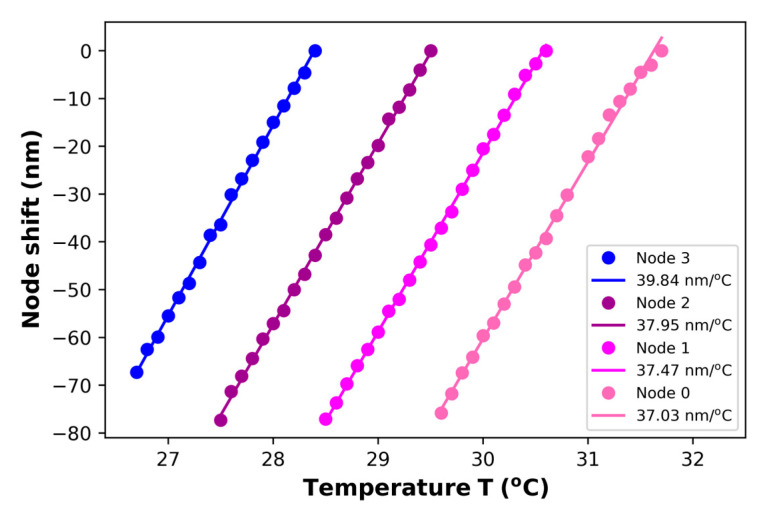
Characteristic curves of the wavelength shift of 4 nodes for a DCFPI using PDMS in a range of temperature of 26.7 to 31.7 °C.

**Figure 12 polymers-15-04567-f012:**
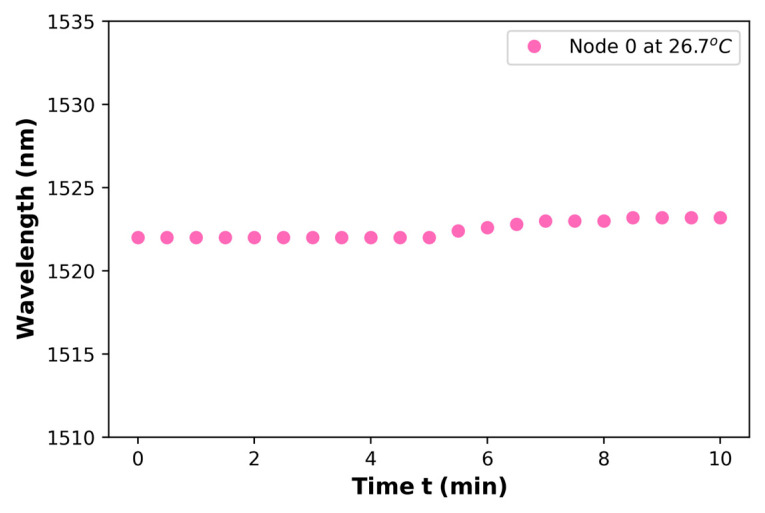
The wavelength shift of node 0 for a DCFPI using PDMS in a constant value of a temperature of 26.7 °C.

**Table 1 polymers-15-04567-t001:** Temperature sensing performance of recently reported temperature optical fiber sensors.

Type	Sensitivity (nm/°C)	Dynamic Range (°C)	Reference
Fiber-optic sensor based on cascaded FPIs	0.18	38–100	[27]
PDMS-filled air microbubble FPI	2.70	51.2–70.5	[28]
Parallel FPIs based on dual Vernier effect	7.61	34–39	[29]
Cascaded FPI and a fixed reflective Lyot filter based on the Vernier effect	−14.63	30–32	[30]
Hybrid interferometers with harmonic Vernier effect	−19.22	41–44	[17]
Cascaded polymer-infiltrated fiber Mach-Zehnder interferometers	−24.86	22–29	[31]
Polymer-capped FFPI by Vernier effect (our work)	39.84	26.7–31.7	

## Data Availability

Data are contained within the article.

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
