# Peer review of "Highly Sensitive Temperature Sensor Based on Vernier Effect Using a Sturdy Double-cavity Fiber Fabry-Perot Interferometer"

_polymers, 2023, doi:10.3390/polym15234567_

Round 1

Reviewer 1 Report

Comments and Suggestions for Authors

1.Please provide more details about how the DCFPI is fabricated and packaged, and how the air cavity length can be adjusted to modify the sensitivity and dynamic range of the sensor.

2.How the coupling coefficients of the interference beams are calculated for the concave polymer surface, and provide some numerical values for comparison.

3.Please provide more details about how the polymer cap thickness is controlled and measured during the fabrication process, and how it affects the sensor performance.

4.It is needed to discuss some potential applications and limitations of the proposed sensor, and suggest some future work directions.

Author Response

We want to thank the reviewers for their positive comments on our work; their suggestions have helped us improve the quality of our manuscript. All the corrections are underlined in the revised manuscript.

Reviewer 2 Report

Comments and Suggestions for Authors

The authors present the design of a DCFPI sensor. Experimental results demonstrate its high sensitivity, achieved through the use of the Vernier effect. Undoubtedly, they deserve attention, but the interpretation leaves many questions

1. For some reason, it is assumed that sensitivity is almost the only characteristic of the sensor. But this is far from the case. Let's assume that I am a potential sensor user. I'm primarily interested in how accurately I can measure temperature and under what conditions. That is, the parameters of interest are accuracy, precision and temperature range.

It is obvious that high sensitivity has a positive effect on the sensor accuracy. But at the same time, high sensitivity leads to the fact that the spectrum quickly shifts "by a full period", that is, a reduction in the permissible measurement range.

The authors know all this very well, but for some reason they do not focus attention on it (it is mentioned in passing in section 3 and - finally! - explicitly in the very last line of the article in the Conclusion)

I think it would be nice to indicate from the very beginning, in the Introduction, that there are various kinds of practical needs for measuring temperature. And among them is to measure temperature in a narrow range with high accuracy (give examples of such applications). And the proposed sensor is ideal for solving precisely such problems. Also add temperature range information to the abstract and to Table 1. So that the potential reader does not have to wonder what all this is for.

2. The authors claim that it is possible to expand the temperature range of measurements by using different peaks in the spectrum.

Please provide a step-by-step algorithm. Again, let's say I'm a potential user of the sensor. As a result of the measurements, I obtained a spectrum (let's say the top one in Figure 8a). I measured the positions of the left and right peaks. But then I need to understand, is it node1 and node2 (in the terminology of the article), or is it node2 and node3, or some others? According to which curve in Figure 11 should I convert the peak position into temperature - blue or pink?

3. Sensitivity is far from the only factor influencing the accuracy and precision of the sensor. The most obvious others are repeatability of results and calibration errors.

Ok, Figure 12 clarifies the question a little about repeatability (although for the convenience of the reader it would be nice to include in the text not only the numbers 12 nm, but also its conversion to temperature - 0.03 ?C)

What about the influence of calibration? I can see by eye the deviations of the experimental data from a straight line in Figures 7-11. Yes, they are not disproportionately large, but you need to estimate what error this will give in ?C.

4. As a minor point.

You always give so many numbers after the decimal point (for example, 39.84 in the abstract). Are you absolutely sure that this is not an overaccuracy? If, for example, you repeat the same experiments, will you get 39.84 again? And not, for example, 39.86? I highly doubt this, given the data in Figure 12.

Comments on the Quality of English Language

English is acceptable

Author Response

(The authors gave the same response as above.)

Round 2

Reviewer 2 Report

Comments and Suggestions for Authors

english is fine